# DSI++: Updating Transformer Memory with New Documents

**Sanket Vaibhav Mehta**[1*]   **Jai Gupta**[2]   **Yi Tay**[3]   **Mostafa Dehghani**[4]   **Vinh Q. Tran**[2]
**Jinfeng Rao**[3]   **Marc Najork**[4]   **Emma Strubell**[1]   **Donald Metzler**[2]

[1]Carnegie Mellon University   [2]Google Research   [3]Google   [4]Google DeepMind

## Abstract

Differentiable Search Indices (DSIs) encode a corpus of documents in model parameters and use the same model to answer user queries directly. Despite the strong performance of DSI models, deploying them in situations where the corpus changes over time is computationally expensive because reindexing the corpus requires re-training the model. In this work, we introduce DSI++, a continual learning challenge for DSI to incrementally index new documents while being able to answer queries related to both previously and newly indexed documents. Across different model scales and document identifier representations, we show that continual indexing of new documents leads to considerable forgetting of previously indexed documents. We also hypothesize and verify that the model experiences forgetting events during training, leading to unstable learning. To mitigate these issues, we investigate two approaches. The first focuses on modifying the training dynamics. Flatter minima implicitly alleviate forgetting, so we optimize for flatter loss basins and show that the model stably memorizes more documents ($+12\%$). Next, we introduce a generative memory to sample pseudo-queries for documents and supplement them during continual indexing to prevent forgetting for the retrieval task. Extensive experiments on novel continual indexing benchmarks based on Natural Questions (NQ) and MS MARCO demonstrate that our proposed solution mitigates forgetting significantly. Concretely, it improves the average Hits@10 by $+21.1\%$ over competitive baselines for NQ and requires 6 times fewer model updates compared to re-training the DSI model for incrementally indexing five corpora in a sequence.

## 1  Introduction

Differentiable Search Indices (DSIs; Tay et al. (2022)) represent a new modeling paradigm for information retrieval tasks using sequence-to-sequence learning. Specifically, DSIs leverage Transformer memory (Vaswani et al., 2017) to encode all of the information in a corpus of documents and then use that memory to answer user queries directly, thereby simplifying the retrieval process. DSIs achieve this functionality by jointly optimizing for indexing (or memorization) and retrieval tasks. The indexing task requires learning a mapping from document content to its identifier, typically represented by integers or short strings (document identifiers, abbreviated *docids*). Then, the retrieval task necessitates mapping user queries to relevant docids. Besides its simplicity and end-to-end differentiable nature, DSI significantly outperforms state-of-the-art "retrieve-and-rank" methods based on dual-encoders (Ni et al., 2022).

Despite the remarkable performance of DSI models, there remain open questions about their applicability in the practical setting of dynamic corpora. Consider the realistic scenario wherein new documents are continually added to the indexed corpus. Updating the index in dual-encoder-based methods requires computing embeddings for new documents, followed by re-indexing all document embeddings (Karpukhin et al., 2020). In contrast, index construction using a DSI involves training a Transformer model. Therefore, the model must be re-trained from scratch every time the underlying corpus is updated, thus incurring prohibitively high computational costs compared to dual-encoders. In this work, we aim to address this issue by devising methods for effective incremental indexing using Transformer memory without re-training the DSI model from scratch.

Lifelong (or continual) learning (Thrun, 1995; Parisi et al., 2019) is a biologically-inspired machine learning paradigm that deals with continuous learning of new tasks by preserving past knowledge and using it to learn new concepts efficiently. Based on this paradigm, we propose *DSI++*

---

* Work performed during an internship at Google Research. Correspondence: sanketvmehta@google.com

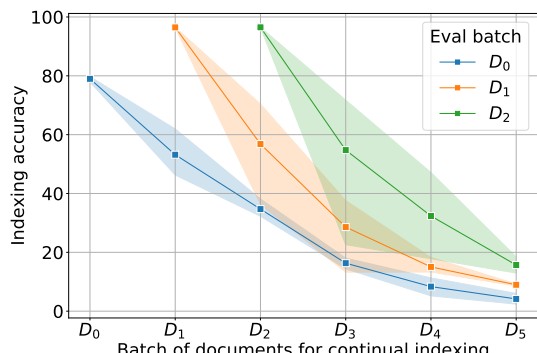

Figure 1: Indexing accuracy of $D_0$, $D_1$, and $D_2$ document corpora visualized as we continuously index new documents (averaged over 3 runs). We observe that continual indexing of new documents leads to severe forgetting of the previously memorized documents.

*(DSI + new documents)*, a continual learning challenge for DSI to incrementally index new documents while maintaining the ability to answer user queries related to both previously and newly indexed documents. To enable DSI++, we introduce novel benchmarks constructed from existing Natural Questions (Kwiatkowski et al., 2019) and MS MARCO (Nguyen et al., 2016) datasets, simulating the continual addition of documents to the system. To our knowledge, there is no prior work studying incremental learning for DSI.

A naive solution for DSI++ is to continuously fine-tune the model with an indexing objective over new documents. However, Figure 1 shows that continual indexing of new documents leads to catastrophic forgetting of the previously memorized documents (more details in §2.1), a common phenomenon in neural networks wherein learning of the new concepts interferes with the previously acquired knowledge (McCloskey and Cohen, 1989). Furthermore, when we investigate the learning dynamics of the DSI model during memorization (Figure 3, we observe a significant number of documents (approx. 88%) experience forgetting events after they have been memorized. Concretely, a *forgetting event* (Toneva et al., 2019) is when a prediction for an individual document goes from correct docid to incorrect one throughout learning. Therefore, *implicit forgetting* during memorization and *explicit forgetting* from continual indexing of new documents are two key challenges to overcome for successfully implementing a DSI++ system.

To reduce forgetting during memorization, we propose explicitly optimizing for flatter loss basins

using Sharpness-Aware Minimization (SAM; Foret et al. (2021)). Recent works have shown that geometrical properties of the minima play a vital role in forgetting, especially models in flatter loss basins tend to undergo less forgetting while lifelong learning from task sequences (Mehta et al., 2023). Next, we introduce a generative memory to sample pseudo-queries for already indexed documents and use them to alleviate forgetting of the retrieval task during incremental indexing of the new documents. Also, the generative memory enables *continual semi-supervised learning* of the retrieval task by generating pseudo-queries for an incoming batch of new documents. Our main contributions can be summarized as follows:

- We introduce DSI++, a continual learning challenge for the recently proposed Differentiable Search Indices (DSI) paradigm. To enable DSI++ evaluations, we create two benchmarks based on existing Natural Questions and MS MARCO datasets. To understand the severity of the forgetting phenomenon across multiple scenarios, we analyze a suite of pretrained models (T5-Base, T5-Large, T5-XL) and different document identifier representations (unstructured atomic, naively structured, and semantically structured).

- We hypothesize and verify that the DSI model experiences forgetting events throughout memorization. To alleviate these, we propose modifying training dynamics to promote flatter minima using SAM and show that the model stably memorizes +12% documents.

- We propose a generative memory-based experience rehearsal approach to alleviate explicit forgetting during continual indexing and improve the average Hits@1 by +25.0% and Hits@10 by +21.1% over competitive baselines for MS MARCO and NQ, respectively.

## 2 DSI++: Continual learning challenge for DSI

### 2.1 Problem setup

We focus on a setup where we receive an initial corpus of documents, $D_0 = \{d_1, \cdots, d_n\}$, and user queries corresponding to a subset of them, $R_0 = \{< q_j, j >, \forall j \in \mathcal{Y}_D\}$, where $D \subset D_0$. DSI paradigm involves two tasks: (i) *memorization task* where the goal is to learn an indexer $f_\theta : \mathcal{X} \rightarrow$

$\mathcal{Y}$, a text-to-text model parameterized by $\theta \in \mathbb{R}^P$, that takes document tokens ($x \in \mathcal{X}$) as input and maps it to a document identifier (docid) $j \in \mathcal{Y}$, and (ii) *retrieval task* where the goal is to use the same indexer $f_\theta$ to directly map a user query $q$ to a relevant docid $j \in \mathcal{Y}$. Two different prompts are used to differentiate between these tasks. Tay et al. (2022) discusses several variants for representing docids – *unstructured atomic* and *structured string* docids, where each document is assigned a unique token and tokenized string, respectively. Under the unified text-to-text format, both of the above tasks are cast as generation tasks, i.e., decoding one unique token (unstructured atomic) or decoding a tokenized string sequentially, one token at a time (naively/ semantically structured).

In the dynamic corpus scenario, we simulate the arrival of new documents by updating the initial corpus $D_0$ with a sequence of batches $D_1 \rightarrow \cdots \rightarrow D_t$. In DSI++, we have access to the new batch of documents $D_i$, but we do not have any queries related to these documents.

**Goal:** Learn a DSI++ system that incrementally indexes $D_1, D_2, \cdots$ in $f_\theta$ while being able to answer queries related to previously as well as additionally indexed documents.

## 2.2 Benchmarks for DSI++

To enable research on DSI++, we introduce two benchmarks constructed from the Natural Questions (NQ; Kwiatkowski et al. (2019)) and MS MARCO (Nguyen et al., 2016) datasets. The NQ dataset consists of Wikipedia articles and corresponding natural language questions. Similar to (Tay et al., 2022), we consider Wikipedia articles for memorization and the retrieval task as identifying the Wikipedia article that answers the given question. We use the original NQ train split to construct train(80%)/ validation(20%) splits and use NQ validation as a test split. We randomly sample $50K$ unique articles to constitute the initial $D_0$ corpus. Next, we construct five corpora ($D_1, \cdots, D_5$), each containing $10K$ unique articles, to add them to the DSI model sequentially. Corresponding to articles in each of these corpora, we filter queries from original NQ train/ validation splits to construct $R_i^{train}, R_i^{val}, R_i^{test}$ ($\forall i \in \{0, \cdots, 5\}$) splits. We use $R_0$ to train the DSI model for the retrieval task and use $R_i^{test}$ to evaluate previously and newly indexed articles. The full MS MARCO dataset has approx. $500K$ passage-query training pairs and

6, 980 validation pairs. Like the benchmark created from the MS MARCO dataset (Pradeep et al., 2023), we randomly sample $50K$ unique passages to constitute the initial $D_0$ corpus and five more corpora, each with $10K$ passages. See Table 2 (in the Appendix) for exact dataset statistics for NQ and MS MARCO.

## 2.3 Evaluation Metrics

For DSI evaluation, we report *indexing accuracy* for memorization task and *Hits@k* ($k \in \{1, 10\}$) metric for retrieval task. Indexing accuracy and Hits@k are the proportion of correctly memorized documents and correct documents ranked in the top k predictions, respectively. We formally define metrics to summarize the model performance as we incrementally index new documents. Let $P_{n,o}$ denote the performance (e.g., indexing accuracy) on corpus $D_o$ after training on corpus $D_n$. Following prior work (Mehta et al., 2023), we compute the **average performance** ($A_n$), **forgetting** ($F_n$) and **learning performance** ($LA_n$) metrics after indexing the corpus $D_n$.

The term $F_n$ (aka *backward transfer*) refers to the effect of indexing the corpus $D_n$ on the performance of all previously indexed documents $D_o$, where $0 \leq o < n$. $LA_n$ (or *forward transfer*) measures the model's ability to learn when presented with a new corpus $D_n$ and is defined as the average performance over the new corpora $D_1, \cdots, D_n$. When the $D_n^{\text{th}}$ corpus is incrementally indexed, $A_n$, $F_n$, and $LA_n$ are defined as follows:

$$A_n = \frac{1}{n+1} \sum_{o=0}^{n} P_{n,o}; \quad LA_n = \frac{1}{n} \sum_{o=1}^{n} P_{o,o};$$

$$F_n = \frac{1}{n} \sum_{o=0}^{n-1} \max_{o' \in \{0, \cdots, n-1\}} (P_{o',o} - P_{n,o}); \quad (1)$$

## 2.4 Case study: Forgetting and Forward Transfer

After introducing the DSI++ problem setup, benchmark, and evaluation metrics, we study the behavior of the DSI model as new documents are continuously added to the system. Concretely, we are interested in investigating the following for continual training of the DSI model with indexing objective on new documents – (Q1) How severe is the forgetting for the initially indexed documents? (Q2) How does continual updating of the DSI model over a sequence of corpora affect the forgetting?

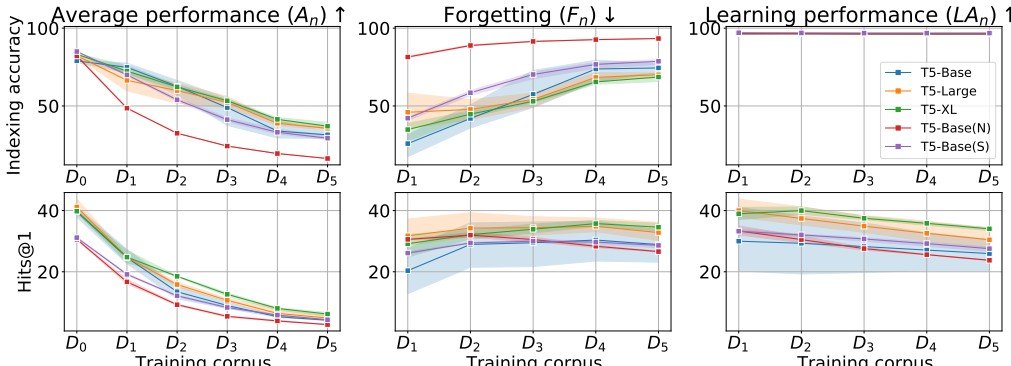

Figure 2: Systematic study about forgetting and forward transfer when incrementally indexing new corpus of documents across different model sizes (T5-Base, T5-Large, T5-XL) and docid representations. We use atomic docids by default and denote (N)/(S) for naively/ semantically structured docids. ↑ indicates higher is better, ↓ indicates lower is better. All results are averaged over 3 runs. We observe that the average $A_n$ and learning $LA_n$ performance improves by increasing the model scale. However, forgetting $F_n$ is severe across all model scales. Next, we observe that naively structured docids, T5-Base(N), underperform unstructured atomic docids, T5-Base, across all metrics - indexing accuracy, Hits@1, (see Figure 6 in Appendix for Hits@10 results). Imbuing the docid space with a semantic (S) structure alleviates the forgetting compared to an arbitrary/ naive (N) structure.

(Q3) How does the updated DSI model perform on newly indexed documents, especially the retrieval task? (Q4) How do different docid representation strategies affect forgetting? (Q5) How does the DSI model scale affect forgetting? Figure 2 visualizes results on the validation split of DSI++ and helps us convincingly answer these questions.

**Forgetting.** From Figure 2, we see that the T5-Base model with atomic docid representation (blue line plots) undergoes significant forgetting. This trend holds across all DSI evaluation metrics - indexing accuracy, Hits@1, and Hits@10 (see 6 in Appendix). For the originally indexed $D_0$ corpus, indexing accuracy and Hits@1 drop by approx. 25 and 20 points, respectively. Further, as we continue indexing the sequence of corpora, we see that forgetting becomes even more severe. For example, after continually indexing the $D_5$ corpus, $F_5$ (forgetting) for indexing accuracy increases to 75. *These results provide evidence to answer (Q1) & (Q2) that the DSI model undergoes severe forgetting under continual indexing of new documents.*

**Forward transfer.** To answer (Q3), we visualize the learning performance ($LA_n$) for all DSI metrics for sequential indexing. From Figure 2, we see $LA_n$ increases in indexing accuracy, suggesting that the DSI model is plastic enough to index new documents. However, from Figure 2, we see a declining trend for Hits@1. Due to the continuous indexing updates, the underlying DSI model

drifts and becomes less effective for the retrieval task. *These findings hint at an approach that replays indexing and retrieval tasks during continual learning (hence our proposed method in §4).*

**Docid representations.** For studying (Q4), we consider unstructured atomic, naively(N) structured, and semantically(S) structured docid representations. From Figure 2, we see that T5-Base(N) underperforms T5-Base by a significant margin. For example, the average performance $A_0$ for the Hits@1 metric is approx. 30 and 39 for naive and atomic docids, respectively. Further, as the naively structured approach treats unstructured docids as tokenizable strings as opposed to dedicated unique tokens in the case of atomic docids, they are relatively more prone to interference from new docids (see $F_n$ subplot for indexing accuracy). *Imbuing semantic structure to the naive docid space helps to reduce forgetting however still underperforms unstructured docids.*

**Model scale.** As atomic docids are superior to naive docids, we only consider atomic docids for answering (Q5). From Figure 2, we observe that larger models outperform their smaller counterparts in terms of the average performance $A_n$ and the learning performance $LA_n$ (T5-XL > T5-Large > T5-Base). *However, empirically we report that forgetting $F_n$ is severe across all model scales, without any clear best performer, and therefore, we focus on T5-Base for the rest of our experiments.*

# 3 Implicit Forgetting: SAM

Memorization (or indexing) is a primary task in the DSI paradigm where the goal is to learn a neural corpus indexer that takes document content as input and maps it to a document identifier (docid). Under the unstructured atomic docid representation strategy, each docid is assigned a unique token/class label. Now given a large number of documents in the corpus (even more than a million), memorization constitutes an instance of challenging extreme classification setting (Bengio et al., 2019). Furthermore, for every class, we have only one labeled example (i.e., document and its identifier), making this task setup rare. Motivated by this largely unexplored setup, we investigate the learning dynamics for the memorization task throughout training.

**Forgetting events.** In Figure 5, we visualize the indexing accuracy for the T5-Base model, optimized with Adafactor (Shazeer and Stern, 2018). We note that the model performance fluctuates throughout training, suggesting unstable memorization. We hypothesize that the model continuously undergoes the forgetting phenomenon wherein subsequent mini-batch updates interfere with the previously memorized documents. To differentiate this phenomenon from forgetting due to adding new documents, we refer to the earlier one as *implicit forgetting* and the latter as *explicit forgetting*. To quantify instability during memorization, we compute forgetting event (Toneva et al., 2019) statistics. *Forgetting event* is defined when an individual document goes from being classified correctly (mapped to correct docid) to incorrectly throughout memorization. In Figure 3, we plot the cumulative histogram of forgetting events where almost 88% of the documents undergo forgetting at least once, validating our hypothesis about implicit forgetting.

**Flatness and forgetting.** Mirzadeh et al. (2020) shows that during sequential learning of tasks, flatter minima leads to less forgetting. Further, Mehta et al. (2023) shows that pre-trained initialization implicitly alleviates forgetting as they prefer flatter minima and explicitly optimizing for the flatness using Sharpness-Aware Minimization (SAM; Foret et al. (2021)) further lessens forgetting. Based on these observations, we hypothesize that modifying the training dynamics of the memorization tasks using SAM should alleviate implicit forgetting.

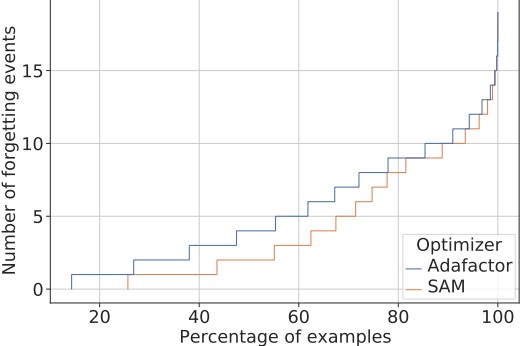

Figure 3: Investigating the effectiveness of SAM for alleviating implicit forgetting in the T5-Base model by visualizing cumulative histogram of forgetting events. A forgetting event (Toneva et al., 2019) is defined when an individual document goes from being classified correctly to incorrectly over the course of memorization. SAM increases the percentage of examples experiencing zero forgetting events by absolute 12% over Adafactor.

**Sharpness-Aware Minimization.** For the loss function $f$, SAM seeks to find the parameters $w$ that lie in the neighborhood with uniformly low loss regions by optimizing the following minimax objective: $\min_w \max_{||\epsilon||_2 \leq \rho} f(w + \epsilon)$, where the maximization region is defined to be a $\ell^p$ ball with radius $\rho$ for $p = 2$. Foret et al. (2021) estimates the gradient of the inner maximization by employing first-order approximation as follows: $\nabla_w \max_{||\epsilon||_2 \leq \rho} f(w + \epsilon) \approx \nabla_w f(w)\big|_{w + \hat{\epsilon}(\mathbf{w})}$, where $\hat{\epsilon}(\mathbf{w}) = \rho \nabla_w f(w)/||\nabla_w f(w)||_2$. For a given mini-batch $B$, SAM approximately computes a point $w' = w + \hat{\epsilon}(w)$ where loss is maximum and then updates the current model weights $w$ using the gradient at $w'$. We defer readers to (Foret et al., 2021) for complete details about this derivation.

**SAM alleviates implicit forgetting.** We investigate the applicability of SAM for alleviating the implicit forgetting phenomenon. We use a pre-trained T5-Base model to memorize $D_0$ corpus containing 50K unique documents. We compare the performance of the SAM with the Adafactor optimizer. In Figure 5, we see that SAM outperforms Adafactor in terms of the overall indexing accuracy. We also note that SAM undergoes less severe fluctuations during training, thus, hinting at less forgetting. To bolster this claim, in Figure 3, we see that SAM has a significantly higher percentage of documents corresponding to a lower cumulative number of forgetting events, i.e., SAM stably (with zero for-

getting events) memorizes $+12\%$ more documents than Adafactor. We also note that SAM ($35.9\pm2.2$) outperforms Adafactor ($32.5\pm6.4$) when evaluated on the retrieval task (Hits@1) corresponding to $D_0$. Therefore, we set SAM to be our default optimizer for the rest of the experiments.

**Discussion.** Mehta et al. (2023) show that explicitly optimizing for flatness using SAM leads to less forgetting, especially in task-incremental learning settings where data undergoes a clear distributional shift. We extend this work to the new DSI paradigm and convincingly demonstrate that SAM helps with the stable memorization of documents. *Our results generalize the earlier findings even to the settings where data does not undergo a clear distributional shift (i.e., memorization task).* Although SAM helps stably memorize documents, there is still room for improvement, and our work invites more future work in this direction.

## 4 Explicit Forgetting: Generative Memory

The DSI paradigm consists of two tasks – memorization and retrieval. The previous section showcases that SAM alleviates implicit forgetting by stably memorizing documents. In this section, we focus on the forgetting phenomenon that arises from the continual indexing of new documents, specifically in the context of the retrieval task. Through our systematic study (in §2.4), we show that irrespective of the model scale and docid representations, DSI models undergo severe forgetting. Moreover, we observe that the learning performance $LA_n$ keeps declining for the retrieval task (see Figures 2 and 6 for Hits@1 and Hits@10, respectively). This observation suggests that as we continuously update the DSI model with the indexing objective, the model forgets the retrieval task. In DSI, both memorization and retrieval tasks return docid for input. By setup, we can assume access to previous documents and continue indexing old and new documents to reduce forgetting of the retrieval task. However, in Figure 4, we see that the model still undergoes forgetting (more in §5.2).

**Episodic memory.** According to the Complementary Learning Systems (McClelland et al., 1995) theory, humans use *episodic memory* to store and revisit past experiences for retaining learned knowledge. Based on this motivation, memory-based approaches (Sodhani et al., 2022), like Experience Replay (ER; Chaudhry et al. (2019)) for continual learning use a subset of previous task data to regularize the future task learning while minimizing forgetting. Based upon this, one approach for DSI++ is to retain ground-truth queries for the retrieval task in episodic memory and use them to co-train with incremental indexing tasks. However, in DSI++, we cannot access ground-truth queries for an incoming batch of new documents. Even if one retains queries for the initial $D_0$ corpus, we show in Table 1 that such a method suffers from forward transfer to newly indexed documents.

**Generative memory.** Recent years have seen significant progress in the capabilities of the generative language models (Raffel et al., 2020; Brown et al., 2020). Motivated by the success of these models and the in-applicability of the episodic memory for DSI++, we pose a question – *instead of retaining the ground-truth queries, can we learn a parametric model to generate such queries given a document*? Concretely, we propose to train a *query generator model* to sample queries for previously seen documents and supplement them during incremental indexing. Since we use the generator model to sample queries for sparse experience replay, our proposed method – *generative memory*. Moreover, generative memory is also used to generate pseudo-queries for the incoming batch of new documents, thus, enabling **continual semi-supervised learning** of the retrieval task.

## 5 Experimentation

In this section, the models are initialized with the pre-trained T5-Base model, while the additional parameters for atomic docid tokens are randomly initialized. See §A.1 for implementation details.

### 5.1 Methods

We compare our proposed generative memory-based approach with the following methods:

**Continual indexing, cl($D_n$).** The DSI model is sequentially fine-tuned with the indexing objective on the incoming corpus of documents $D_n$.

**Continual indexing with all seen documents, cl($U_n$).** The DSI model is continuously fine-tuned with the indexing objective on the updated corpora $U_n$ ($\bigcup_{i=0}^{n} D_i$) with the same replay frequency for the old ($\bigcup_{i=0}^{n-1} D_i$) and new ($D_n$) corpora in the tasks mixture.

| Added corpus | Method | Eval corpus = $D_0$ (Catastrophic forgetting) | | | Eval corpus = $D_1$ (Forward transfer) | | |
|---|---|---|---|---|---|---|---|
| | | Index acc. | Hits@1 | Hits@10 | Index acc. | Hits@1 | Hits@10 |
| $D_0$ | - | $81.8_{1.2}$ | $35.9_{2.2}$ | $66.9_{0.9}$ | - | - | - |
| $D_1$ | cl($D_1$) | $52.4_{3.5}$ | $19.2_{3.9}$ | $43.6_{5.7}$ | $96.5_{0.0}$ | $31.7_{6.4}$ | $55.6_{4.9}$ |
| | cl($U_1 = D_0 \cup D_1$) | $78.2_{0.5}$ | $28.9_{8.9}$ | $59.0_{7.9}$ | $91.8_{0.4}$ | $34.0_{2.4}$ | $60.2_{1.9}$ |
| | cl($U_1$)+epsmem($D_0$) | $77.8_{0.5}$ | $22.9_{1.5}$ | $51.4_{0.5}$ | $93.1_{0.0}$ | $13.1_{2.1}$ | $39.6_{3.1}$ |
| | cl($U_1$)+genmem($D_0$) | $77.8_{0.3}$ | $26.0_{6.9}$ | $54.9_{8.3}$ | $93.0_{0.5}$ | $8.6_{4.8}$ | $31.6_{11.8}$ |
| | cl($U_1$)+epsmem($D_1$) | $53.2_{3.1}$ | $7.7_{2.1}$ | $26.0_{2.0}$ | $96.5_{0.0}$ | $48.3_{2.3}$ | $70.7_{1.9}$ |
| | cl($U_1$)+genmem($D_1$) | $50.1_{0.8}$ | $7.0_{1.2}$ | $23.1_{2.2}$ | $96.5_{0.0}$ | $57.7_{1.5}$ | $76.7_{0.9}$ |
| | cl($U_1$)+genmem($U_1$) | $78.2_{0.3}$ | $18.4_{2.8}$ | $47.5_{3.9}$ | $92.1_{0.3}$ | $48.5_{6.1}$ | $73.8_{2.9}$ |
| | cl($U_1$, docid parameters only) | $78.9_{0.1}$ | $32.7_{5.1}$ | $64.8_{4.2}$ | $94.6_{0.1}$ | $10.8_{3.8}$ | $35.0_{7.3}$ |
| | train from scratch | $78.7_{0.6}$ | $35.9_{1.4}$ | $66.4_{0.0}$ | $79.2_{0.3}$ | $32.9_{1.8}$ | $63.9_{1.2}$ |

Table 1: Comparing performance on incremental indexing of $D_1$ corpus across different methods - cl($D_1$): continue fine-tuning with indexing task on $D_1$, cl($U_1$): continue fine-tuning on the updated corpus $U_1$, cl($U_1$)+epsmem($D$): continual indexing of $U_1$ along with ER of queries for $D$, cl($U_1$)+genmem($D$): continual indexing of $U_1$ along with ER of pseudo-queries for $D$. We observe that continual indexing on the updated corpus cl($U_1$) reduces forgetting compared to just indexing new corpus cl($D_1$) in the Natural Questions (NQ) dataset ($|D_0| = 50K$, $|D_1| = 10K$). Next, ER with either $D_0$ or $D_1$ hurts forward transfer or forgetting. Our proposed approach of augmenting pseudo-queries for all documents along with continual indexing, cl($U_1$)+genmem($U_1$), alleviates forgetting of $D_0$ corpus and improves forward transfer to $D_1$ corpus.

**Continual experience replay using generative memory, genmem($D_n$).** In this method, the proposed generative memory model is used to sample pseudo-queries corresponding to the corpus $D_n$. Next, these pseudo-queries are used for (sparse) experience replay of the retrieval task samples.

**Continual experience replay using episodic memory, epsmem($D_n$).** In this method, ground-truth queries corresponding to the $D_n^{th}$ corpus are used for experience replay of the retrieval task.

**cl($U_n$, docid parameters only).** In this method, we only update the parameters corresponding to atomic docid tokens using the updated $U_n$ corpus. This method in spirit is a dual-encoder-baseline.

**Train from scratch, (no cl).** The DSI model is trained from scratch every time a new corpus is added. This method corresponds to a non-continual learning setup and is computationally expensive.

## 5.2 Results

In this section, we revisit some of the questions (Q1)-(Q3) raised in our case study (see §2.4) to investigate the effectiveness of our proposed generative memory-based approach. To answer these questions, in Table 1, we report the performance of the DSI model on $D_0$ (to study the forgetting phenomenon) and $D_1$ corpora (to answer forward transfer question) after continual indexing on $D_1$ for both NQ and MS MARCO datasets. In Figures 4 and 7 (NQ) and Figure 8 (MS MARCO), we report overall performance across DSI metrics as we continuously update the model with the sequence of five corpora ($D_1 \rightarrow \cdots \rightarrow D_5$).

**Does generative memory alleviate forgetting of old documents?** In Table 1, for the NQ dataset, we report Hits@1 to be 35.9 for the model after training on $D_0$. We see that continually indexing both $D_0$ and $D_1$ corpora (cl($U_1$) - 28.9), significantly reduce forgetting the retrieval task (Hits@1) over just indexing the new corpora $D_1$ (cl($D_1$) - 19.2). Next, we look at the performance of the ER approaches when augmented with the continual indexing of all documents. We see that both episodic memory (cl($U_1$)+epsmem($D_0$) - 22.9), and generative memory (cl($U_1$)+genmem($D_0$) - 26.0) reduce forgetting compared to cl($D_1$) when we replay (pseudo-)queries corresponding to $D_0$ corpus. Moreover, generative memory outperforms episodic memory without retaining original queries. Although from Table 1, we see generative memory, cl($U_1$)+genmem($U_1$), underperforms cl($U_1$), from Figures 4 and 7, we see that generative memory, cl($U_5$)+genmem($U_5$), outperforms cl($U_5$) both in terms of average performance $A_n$ and forgetting $F_n$ over five sequential updates. *These results convincingly show that the ER with generative memory significantly alleviates forgetting the retrieval task compared to considered baselines.*

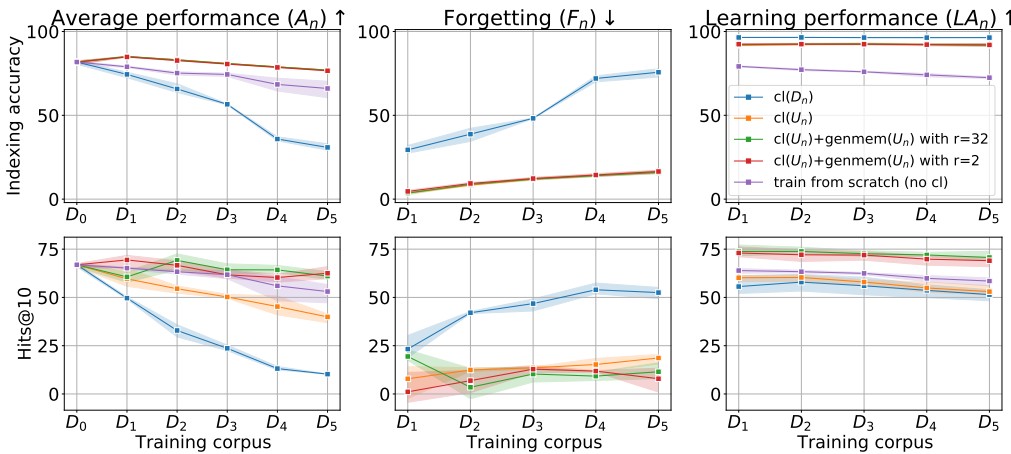

Figure 4: Investigating the effectiveness of generative memory in mitigating forgetting when continuously indexing new corpus $D_n$ (T5-Base model and atomic docids representation) for the NQ dataset. ↑ indicates higher is better, ↓ indicates lower is better. We observe that continual indexing of old and new documents $cl(U_n)$ helps to alleviate forgetting of older documents when evaluated on retrieval tasks. However, average Hits@10 ($A_n$) still undergo 23 points drop after sequential updates ($D_0 \rightarrow D_1 \cdots \rightarrow D_5$). Generative memory enables sparse replaying of pseudo-queries for old documents and continual semi-supervised learning with new documents. We observe that augmenting generative memory during continual indexing not only reduces the forgetting ($F_n$) but also improves average Hits@10 ($A_n$) by $+21.1\%$ over considered baselines (see Figure 7 for Hits@1 results. Figure 8 for MS MARCO results in the Appendix).

**Does generative memory enable forward transfer to new documents?** One of the goals of DSI++ is to enable answering queries related to newly indexed documents. Towards this goal, in Table 1, for the NQ dataset, we look at the retrieval task performance (Hits@1) for $D_1$ after incrementally indexing $D_1$. To compare different methods, we consider a baseline in the form of ER with ground-truth queries for $D_1$ ($cl(U_1)+epsmem(D_1)$ - 48.3). We see that without any fine-tuning on the retrieval task for $D_1$, incremental learning with indexing objective shows impressive forward transfer (or zero-shot gains, $cl(D_1)$ - 31.7 and $cl(U_1)$ - 34.0). Moreover, ER with generative memory outperforms supervised baseline ($cl(U_1)+genmem(D_1)$ - 57.7). However, we notice that replaying queries corresponding to either $D_0$ or $D_1$ hurt forward transfer to $D_1$ ($cl(U_1)+genmem(D_0)$ - 8.6) or amplify forgetting of $D_0$ ($cl(U_1)+genmem(D_1)$ - 7.0). These results suggest that the memory module should include (pseudo-)queries corresponding to old and new documents. From Figure 4, we see that continual indexing method $cl(U_n)$ has a downward trend for $LA_n$ (Hits@10), therefore, eventually forgetting the retrieval task. On the other hand, ER with generative memory is relatively constant, providing evidence against forgetting. *In summary, ER*

*with generative memory enhances retrieval task performance by reducing forgetting of indexed documents and enabling forward transfer to newly indexed documents.*

**Does generative memory generalize to different datasets?** In Table 3, for the MS MARCO dataset, we report Hits@1 to be 78.2 after training on $D_0$ passages. We see that continually indexing both $D_0$ and $D_1$ corpora ($cl(U_1)$ - 76.5 and $cl(U_1)+genmem(U_1)$ - 73.7), significantly reduce forgetting the retrieval task (Hits@1) over just indexing the new corpora $D_1$ ($cl(D_1)$ - 68.0). Next, we look at the retrieval task performance (Hits@1) for $D_1$ after incrementally indexing $D_1$. We see that without any fine-tuning on the retrieval task for $D_1$, incremental learning with indexing objective shows impressive forward transfer ($cl(D_1)$ - 36.1 and $cl(U_1)$ - 35.3). Moreover, ER with generative memory, $cl(U_1)+genmem(U_1)$ - 80.6, performs far superior to just incremental indexing objective. Similar to the results with the NQ dataset, we show that ER with generative memory, $cl(U_n)+genmem(U_n)$, improves the overall performance for the retrieval task, reducing forgetting of previously indexed documents and enables forward transfer to new documents compared to continual indexing of all documents, $cl(U_n)$. *We show that*

*our results hold across two datasets, thus, show-casing the generalizability of our approach.*

**Investigating the effectiveness of the generative memory with the scale of a corpus.** We conduct experiments with a full MS MARCO dataset ($\approx 8.9M$ passages). We construct two corpora $- D_0 = 8M$ and $D_1 = 841,823$ passages. We train the DSI model using $D_0$ passages and incremental add $D_1$ passages. In Table 3, we report results for MS MARCO. We see that continual fine-tuning with the indexing task on $D_1$, cl($D_1$), completely forgets the retrieval task for $D_0$ passages (Hits@1 goes to 0.1 from 16.3). However, the generative memory-based approach significantly reduces forgetting (Hits@1 of 7.3). Moreover, generative memory enables continual semi-supervised learning by augmenting pseudo-queries for $D_1$ passages, thereby improving forward transfer (Hits@1 of 31.6 vs. 18.2 for cl($D_1$)). *Our proposed solution reduces forgetting in large corpus settings.*

**Investigating sparsity of experience replay (ER) on forgetting.** ER with generative memory co-trains the indexing and pseudo-labeled retrieval tasks. Tay et al. (2022) introduces a mixing ratio $r$ to define the ratio of indexing to retrieval samples. The mixing ratio is inversely related to the sparsity of ER, i.e., higher $r$ (more indexing samples) corresponds to sparse updates from pseudo-labeled retrieval samples. Following (Tay et al., 2022), we consider $r = \{2, 32\}$ for our analysis. From Figure 4, we see that $r = 32$ (sparse replay) slightly outperforms $r = 2$ in terms of average performance, forgetting, and learning accuracy. *These results suggest that even sparse regularization updates from ER positively influence backward and forward transfer in DSI++.*

**Analyzing index construction time for DSI++.** DSI involves training a Transformer model for index construction. DSI++ allows incremental updating of the indexer. In Figures 4, 7, and 8, we demonstrate that our incremental indexer updating method surpasses the "train from scratch" baseline in terms of $A_n$. Note that the "train from scratch" baseline can serve as a performance upper bound for continual learning when there is no detrimental interference among tasks, and all tasks are evenly balanced. However, in the case of DSI++, there exists an initial base corpus that is larger than subsequent corpora, leading to an imbalance among tasks. Consequently, "train from scratch" should

be regarded as a competitive baseline rather than an inherent upper bound. This is also the reason behind reporting the learning accuracy ($LA_n$) for every metric, which can be seen as an upper bound since it maintains a running average of the best performance across all corpora. Furthermore, one of the key objectives of continual learning is to leverage prior knowledge to enhance the learning of new tasks. Indeed, from Tables 1 and 3, we observe that our proposed method excels in forward transfer compared to the "train from scratch" approach.

For the NQ dataset, indexing the initial $D_0$ corpus of 50K documents requires 350K training steps. If we sequentially index additional $D_1$ to $D_5$ corpora (10K each) by re-training the DSI model each time, it would require around 1.75M steps. In contrast, our approach only requires slightly above 300K additional updates to incrementally index all corpora, which is approximately six times fewer updates. *Our approach achieves superior overall performance compared to re-training from scratch, while also being more computationally efficient.*

## 6 Conclusion

DSI++ introduces a new approach to address a crucial requirement of DSI models for practical use in production setups, where continuous addition of new documents to the corpus is necessary. Through experiments, we demonstrate the effectiveness of our proposed solutions: sharpness-aware minimization and generative memory, which significantly reduce catastrophic forgetting. This work establishes a foundation for further research, benefiting both DSI models and the broader community of continual (semi-supervised) learning.

## Limitations

In this study, we explore the phenomenon of forgetting in relation to the addition of *new and distinct* documents into the indexer. It is important to note that when a new document refutes or modifies a previously indexed document, the model's behavior becomes unpredictable, requiring further analysis. Additionally, we examine the effectiveness of our proposed method on a larger dataset, such as the full MS MARCO dataset. However, it is worth noting that with this larger dataset, the method exhibits significant forgetting. As a result, additional research is necessary to enhance the model's performance, particularly when dealing with datasets of larger scales.

## Ethics Statement

Training large models is expensive and can have a detrimental impact on the environment (Strubell et al., 2019). Continual learning on top of existing models is preferable to re-training from scratch in this regard since it requires many fewer training steps. With DSI++, we aim to reduce the need to re-train DSI models from scratch whenever a new set of documents is added to the corpus thereby making it cheaper and better for the environment. Concretely, in §5.2, we analyze the index construction time for DSI++ and show that our approach is computationally efficient in comparison to re-training the model from scratch. At the same time, we acknowledge that reduced cost can increase overall consumption (Jevons' paradox).

## Acknowledgements

We thank the anonymous reviewers for their valuable feedback and suggestions, which helped improve the paper. We also thank Ronak Pradeep and Kai Hui for help with the MS MARCO setup, Tal Schuster and Raghuram Mandyam Annasamy for reviewing the paper, and William W. Cohen, Aditya Gupta, Dara Bahri, and Fuzhao Xue for sharing insights and intuitions during initial discussions. We would like to thank COMEDY (COhorts of Maarten Sap, Emma Strubell, Daniel Fried, and Yonatan Bisk) lab members for reviewing the paper and providing valuable comments; Jeremiah Milbauer, Clara Na, Jared Fernandez, Nupoor Gandhi, Zhisong Zhang, and Vijay Viswanathan also gave constructive feedback on drafts and tables.

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

# A   Appendix

## A.1   Implementation Details

We utilize the pre-trained T5-Base (Raffel et al., 2020) to initialize all models and randomly initialize the additional parameters for atomic docid tokens. Bahri et al. (2022) demonstrates the successful applicability of SAM for language model generalization, especially in pre-trained T5 models. We mainly follow (Bahri et al., 2022) to set our hyper-parameters: $\rho = 0.15$, batch size=32 for the inner maximization step in SAM.

While indexing $D_0$ corpus, we train all the models for a maximum of 1M steps with a warmup of 100K steps. During continual indexing of other corpora, we train for a maximum of 100K steps with a warmup of 100 steps. For the rest of the hyperparameters, we follow (Tay et al., 2022) – set a learning rate to 0.001, batch size to 128, and input sequence length to 32. We evaluate models after every 5K steps and retain the checkpoint yielding the best performance. For the initial training with $D_0$ corpus, we co-train on indexing and retrieval tasks; therefore, we use the average of all DSI metrics (indexing accuracy, Hits@1, and Hits@10) for model

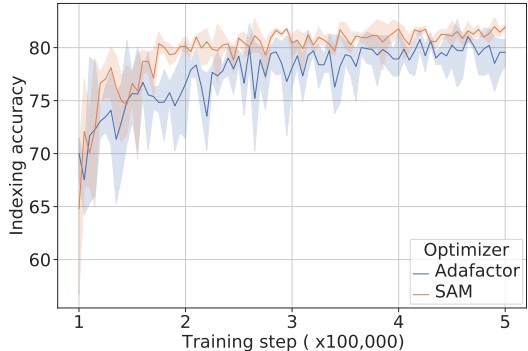

Figure 5: Investigating the effectiveness of SAM for alleviating implicit forgetting in the T5-Base model by visualizing indexing accuracy during memorization. We observe serious fluctuations in the indexing accuracy in the case of the Adafactor optimizer, thereby suggesting unstable memorization. SAM leads to relatively stable memorization of documents.

selection. For the continual learning experiments, we have access to only indexing accuracy for all involved corpora, so we use it for model selection.

To train a parametric model for generative memory, we utilize the retrieval dataset $R_0$, which corresponds to the $D_0$ corpus. We set the maximum sequence length for document contents to 1024, the target length for generated queries to 32, batch size to 128, train for a maximum of 100K steps, and use BLUE for model selection. We use beam decoding to generate pseudo-queries. We tune the learning rate amongst $\{0.001, 0.0005\}$ and linear warmup amongst $\{1K, 10K\}$. For all our experiments, we use the T5X (Roberts et al., 2022) framework along with 4-8 TPUv4 chips to train the models.

## A.2 Related Work

We review relevant prior work along two dimensions: Application setups related to DSI++ and continual learning methods to alleviate forgetting and enable forward transfer.

**Language models (LMs) as knowledge bases (KBs).** Petroni et al. (2019) shows that pre-trained BERT (Devlin et al., 2019) models capture relational knowledge comparable to that of the KBs constructed using off-the-shelf techniques. Concretely, these models can be used to extract factual knowledge about relations between entities by providing a prompt to predict missing words in a cloze-style template (e.g., "New Delhi is the capital of __"). Similarly, Roberts et al. (2020) demon-

strates that pre-trained T5 (Raffel et al., 2020) models can be employed to answer open-domain questions without access to any external knowledge or context. However, unlike structured KBs, it is non-trivial to update knowledge stored implicitly in the weights of these models. Therefore, Zhu et al. (2020) introduces an experimentation setup where the task is to update facts stored within the pre-trained models and proposes a constrained optimization method, similar to Elastic Weight Consolidation (Kirkpatrick et al., 2017), to alleviate catastrophic forgetting. With similar motivation, (Dhingra et al., 2022) introduces a diagnostic dataset to probe LMs for facts that change over time. It also suggests jointly modeling text with its timestamp for improved memorization of seen facts. Recent works have been investigating efficient ways to localize and edit facts stored with the LMs (AlKhamissi et al., 2022) using finetuning (Zhu et al., 2020; Dhingra et al., 2022), hyper-networks (De Cao et al., 2021; Mitchell et al., 2022), and direct editing (Meng et al., 2022). Although a crucial line of work around updating facts in the pre-trained LMs, using prompting as our probing mechanism only provides a lower bound estimate of the knowledge contained in these models (Jiang et al., 2020). On the other hand, we explicitly focus on the memorization task in DSI++. This task helps us to answer questions related to catastrophic forgetting more convincingly rather than bounded by the mechanism of how we probe these models.

**Optimization-based approaches** for continual learning encode the necessary inductive biases required to enable continual learning by modifying the training dynamics. Flatter minima are shown to alleviate forgetting (Mirzadeh et al., 2020). Further, Mehta et al. (2023) showed that explicitly optimizing for flatter loss basins using Sharpness-Aware Minimization (SAM; Foret et al. (2021)) reduces forgetting. Building on these works, we show that flatter minima induced by SAM reduce implicit forgetting during memorization, thereby leading to more stable memorization (see §3).

**Memory-based (aka data-based regularization) approaches** for continual learning constrain the parameter updates based on the previous task examples sampled from memory. Sparse experience replay using episodic memory (Chaudhry et al., 2019) is a prominent approach, and in §4, we discuss its limitations of it for DSI++. Next, Shin et al.

| Dataset | #D | Natural Questions (NQ) | | | MS MARCO | | |
|---|---|---|---|---|---|---|---|
| | | #Train | #Validation | #Test | #Train | #Validation | #Test |
| $R_0$ | 50K | 53.8K | 13.5K | 3.9K | 2M | 25.0K | 3.6K |
| $R_1$ | 10K | 10.7K | 2.7K | 809 | 400K | 5.1K | 762 |
| $R_2$ | 10K | 10.6K | 2.7K | 787 | 400K | 5.1K | 770 |
| $R_3$ | 10K | 10.7K | 2.7K | 727 | 400K | 4.9K | 734 |
| $R_4$ | 10K | 10.9K | 2.7K | 772 | 400K | 4.9K | 730 |
| $R_5$ | 10K | 10.7K | 2.7K | 847 | 400K | 4.9K | 660 |

Table 2: DSI++ dataset statistics for NQ and MS MARCO: memorization and retrieval tasks.

| Added corpus | Method | Eval corpus = $D_0$ (Catastrophic forgetting) | | | Eval corpus = $D_1$ (Forward transfer) | | |
|---|---|---|---|---|---|---|---|
| | | Index acc. | Hits@1 | Hits@10 | Index acc. | Hits@1 | Hits@10 |
| *MS MARCO $-  |D_0| = 50K, |D_1| = 10K$* | | | | | | | |
| $D_0$ | - | $99.4_{0.2}$ | $78.2_{0.2}$ | $95.0_{0.1}$ | - | - | - |
| $D_1$ | cl($D_1$) | $46.7_{18.6}$ | $68.0_{2.0}$ | $87.3_{1.3}$ | $99.8_{0.0}$ | $36.1_{9.5}$ | $65.8_{6.9}$ |
| | cl($U_1$) | $99.4_{0.0}$ | $76.5_{0.7}$ | $94.2_{0.3}$ | $99.8_{0.0}$ | $35.3_{4.1}$ | $64.4_{3.3}$ |
| | cl($U_1$)+genmem($U_1$) | $99.3_{0.1}$ | $73.7_{0.2}$ | $93.9_{0.3}$ | $99.8_{0.0}$ | $80.6_{1.0}$ | $95.5_{0.1}$ |
| | train from scratch | $99.5_{0.0}$ | $75.0_{0.2}$ | $93.9_{0.1}$ | $99.6_{0.0}$ | $73.4_{1.3}$ | $93.4_{0.9}$ |
| *MS MARCO (full) $- |D_0| = 8M, |D_1| = 842K$* | | | | | | | |
| $D_0$ | - | 99.4 | 16.3 | 46.8 | - | - | - |
| $D_1$ | cl($D_1$) | 0.0 | 0.1 | 0.6 | 97.9 | 18.2 | 40.5 |
| | cl($U_1$)+genmem($U_1$) | 20.4 | 7.3 | 31.3 | 86.6 | 31.6 | 65.8 |

Table 3: Comparing performance on incremental indexing of $D_1$ corpus across different methods - cl($D_1$): continue fine-tuning with indexing task on $D_1$, cl($U_1$): continue fine-tuning on the updated corpus $U_1$, cl($U_1$)+genmem($D$): continual indexing of $U_1$ along with ER of pseudo-queries for $D$. We observe that continual indexing on the updated corpus cl($U_1$) reduces forgetting compared to just indexing new corpus cl($D_1$) in the MS MARCO dataset. Our proposed approach of augmenting pseudo-queries for all documents along with continual indexing, cl($U_1$)+genmem($U_1$), alleviates forgetting of $D_0$ corpus and improves forward transfer to $D_1$ corpus. We also show that our proposed solution reduces forgetting of $D_0(= 8M)$ passages while incremental indexing in a large corpus setting, MS MARCO (full) containing $8.9M$ passages.

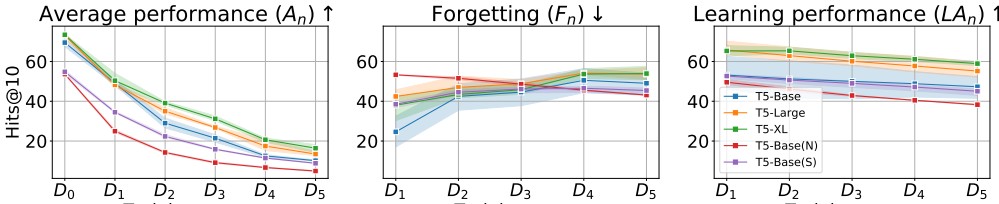

Figure 6: Systematic study about forgetting and forward transfer when incrementally indexing new corpus of documents across different model sizes (T5-Base, T5-Large, T5-XL) and docid representations. We use atomic docids by default and denote (N)/(S) for naively/semantically structured string docids. ↑ indicates higher is better, ↓ indicates lower is better. We observe that by increasing the model scale, the average $A_n$ and learning $LA_n$ performance improves. However, forgetting $F_n$ is severe across all model scales. Moreover, we observe that naive string docids (N) underperform atomic docids across the Hits@10 metric. Similar to Figure 2, imbuing the docid space with a semantic (S) structure alleviates the forgetting compared to an arbitrary/ naive (N) structure.

(2017); Sun et al. (2020) learns a parametric model to reconstruct the examples for seen tasks. However, in DSI++, we do not see queries for the new documents. Therefore, we use a parametric memory to generate pseudo-queries for already indexed (older) documents and an incoming batch of new documents, thus, enabling us to leverage unlabeled data (in the form of new documents) for continual semi-supervised learning. On the other hand, Sun et al. (2020) assumes that the incoming data are

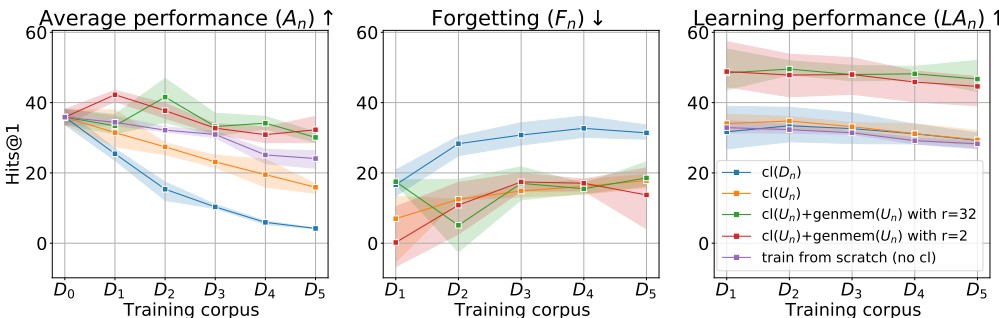

Figure 7: Investigating the effectiveness of generative memory in mitigating forgetting when continuously indexing new corpus $D_n$ (T5-Base model and atomic docids representation) for the NQ dataset. ↑ indicates higher is better, ↓ indicates lower is better. We observe that continual indexing of old and new documents cl($U_n$) helps to alleviate forgetting of older documents when evaluated on retrieval tasks. However, average Hits@1 ($A_n$) still undergo 19 points drop after sequential updates ($D_0 \rightarrow D_1 \cdots \rightarrow D_5$). We observe that augmenting generative memory during continual indexing not only reduces the forgetting ($F_n$) but also improves average Hits@1 ($A_n$) by +17.3% over continual indexing.

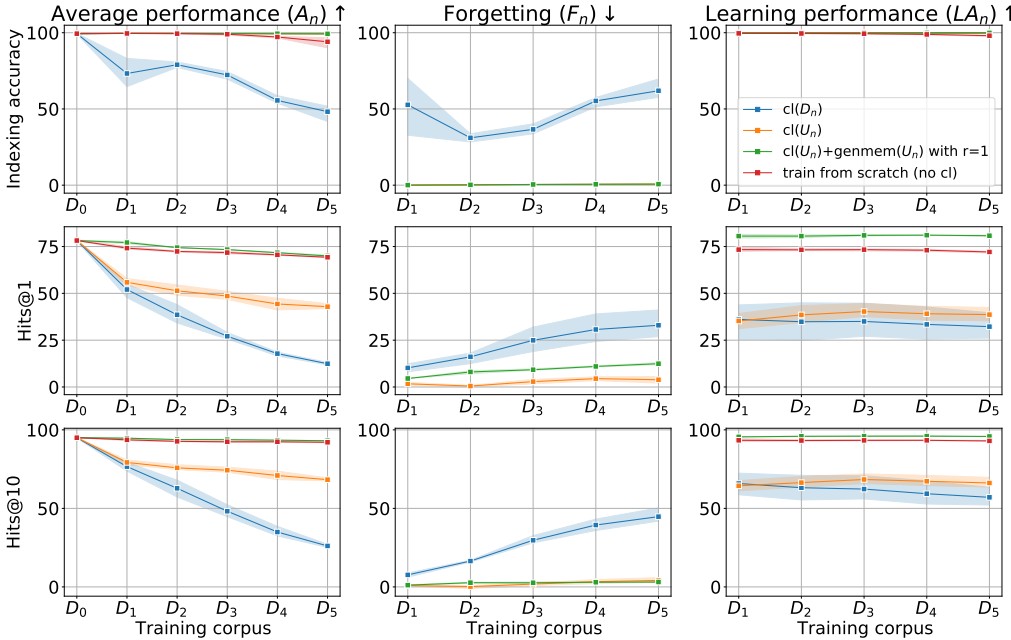

Figure 8: Investigating the effectiveness of generative memory in mitigating forgetting when continuously indexing new corpus $D_n$ (T5-Base model and atomic docids representation) for the MS MARCO dataset. ↑ indicates higher is better, ↓ indicates lower is better. We observe that continual indexing of old and new documents cl($U_n$) helps to alleviate forgetting of older documents when evaluated on retrieval tasks. However, average Hits@10 ($A_n$) still undergo 25.0 points drop after sequential updates ($D_0 \rightarrow D_1 \cdots \rightarrow D_5$). Generative memory enables sparse replaying of pseudo-queries for old documents and continual semi-supervised learning with new documents. We observe that augmenting generative memory during continual indexing not only reduces the forgetting ($F_n$) but also improves average Hits@10 ($A_n$) by +23.0% over considered baselines.

fully labeled, which is not applicable in DSI++ (we do not get to see queries for the new documents). Furthermore, Sun et al. (2020) shows that using a parametric model underperforms episodic mem-

ory. In our work, we do not generate example pairs $(x, y)$ but rather generate pseudo-queries $(y)$, similar to contemporary works (Zhuang et al., 2022; Bonifacio et al., 2022). We show that our approach

outperforms episodic memory. Lastly, in the context of pseudo-query generation, neural models are prone to hallucinate additional content not supported by the input documents. Future works can study methods to filter out noisy pseudo-queries (Mehta et al., 2022) during incremental indexing.

**Test time adaptation approaches** for continual learning use episodic memory at the inference time to alter the model weights before making predictions (Rebuffi et al., 2017; Sprechmann et al., 2018; de Masson D'Autume et al., 2019; Wang et al., 2020). Updating the DSI indexer for every user query is computationally expensive, so we focus on continual learning methods during training. Apart from continual learning-focused approaches, retrieval augmented generation (Guu et al., 2020; Izacard and Grave, 2021; Borgeaud et al., 2022) family of approaches retrieve auxiliary passages/documents to enhance pre-trained language models. These approaches alter test-time predictions of the generative models by augmenting their input with relevant passages retrieved from external retrievable memory. Moreover, one explicitly disables the updates to the employed pre-trained (and retrieval) model using the external retrievable memory. Such approaches do not faithfully assess the fundamental challenge of learning continually, specifically catastrophic forgetting. On the other hand, our work focuses on the recently introduced DSI paradigm (Tay et al., 2022), where information in the document corpus is encoded into the model parameters. Therefore, any updates to the underlying corpus necessitate updating the model parameters hence, undergoing severe forgetting. Our work tackles a more challenging setup for studying the forgetting phenomenon in detail. However, retrieval-augmented generation-based methods do not analyze the forgetting phenomenon, only looking at overall performance metrics. We agree that continual learning is broader than catastrophic forgetting. However, in this work, we decided to study the forgetting phenomenon in detail on one of the most challenging setups, if not the most difficult.

**Parameter isolation-based approaches** for continual learning assign different dedicated subsets of the model parameters to each task to prevent forgetting (De Lange et al., 2021). While learning a new task, these methods either freeze a subset of the parameters corresponding to older tasks or dynamically add new parameters per new task. At the prediction time, these methods typically require task identity to activate the corresponding subset of parameters for inference. In the DSI paradigm, we are given user queries at the inference time, and the goal is to predict relevant document identifiers. Now during incremental indexing, if we consider every new document corpus as a new task, then a typical parameter isolation-based approach would require corpus identity for every user query at the test time, defeating the whole purpose of the DSI paradigm. Due to this, the parameter isolation-based approaches in their current form are rendered less useful for DSI++. Nevertheless, we believe that by masking the weights for the already indexed corpus, one is explicitly disabling the updates to the underlying DSI model; therefore, parameter isolation-based methods would be robust to forgetting, and future works should explore them for DSI++. We believe, however, that adapting these methods for DSI++ is out of scope for this paper, and we would not be able to do both this topic and our current work justice in the limited space available.