# OpenReview forum: "DSI++: Updating Transformer Memory with New Documents"
_EMNLP/2023/Conference — EMNLP 2023 Main_

### Official Review · Reviewer_Gobp · 2023-08-04

**Soundness:** 3

**Excitement:**

4: Strong: This paper deepens the understanding of some phenomenon or lowers the barriers to an existing research direction.

**Paper Topic And Main Contributions:**

The paper proposes to extend the existing SDI model which can incrementally index new documents with a lower computational cost.

**Reasons To Accept:**

1. The motivations of some designs are clear.
2. The paper is detailed in the steps and well written.

**Reasons To Reject:**

1. Implementation details and related work should be included in the content rather than the appendix.
2. The source code for training the proposed model is not released. It would be useful for ongoing research efforts if the paper could release its source code and predicted results.

**Reproducibility:**

3: Could reproduce the results with some difficulty. The settings of parameters are underspecified or subjectively determined; the training/evaluation data are not widely available.

**Reviewer Confidence:**

3: Pretty sure, but there's a chance I missed something. Although I have a good feel for this area in general, I did not carefully check the paper's details, e.g., the math, experimental design, or novelty.

---

> ### Author Rebuttal · Authors · 2023-08-29
>
> We thank the reviewer for their time and remarks on our work. We are pleased that the reviewer appreciates the **strong motivation, clarity, and quality of writing** in our paper!
>
> **[Implementation details and related work in appendix]**
>
> We acknowledge the reviewer's concern regarding the inclusion of implementation details and related work in the appendix. However, **we want to emphasize that we have added relevant discussions of related work throughout the main text whenever necessary**. For instance, we delve into optimization-based approaches for continual learning in lines 337-347 and discuss episodic memory-based methods in lines 418-435. Nevertheless, in response to the reviewer's suggestion, we are committed to enhancing the comprehensiveness of our paper. **With the additional page available in the camera-ready version, we will incorporate the details currently found in the appendix into the main text for easier access and reference**.
>
> **[Source code and results]**
>
> In our reproducibility statement (lines 62-674), we have clearly mentioned that **upon publication, we plan to release our code and dataset splits to reproduce our experiments and actively support ongoing research endeavors related to DSI++**. Also, in the appendix, we will tabulate exact numeric values for plots visualized in Figures 4, 6, 7, and 8.
>
> **We once again thank the reviewer for their positive feedback on our manuscript and look forward to answering any other questions!**

---

### Official Review · Reviewer_mZs9 · 2023-08-05

**Soundness:** 3

**Excitement:**

3: Ambivalent: It has merits (e.g., it reports state-of-the-art results, the idea is nice), but there are key weaknesses (e.g., it describes incremental work), and it can significantly benefit from another round of revision. However, I won't object to accepting it if my co-reviewers champion it.

**Paper Topic And Main Contributions:**

This paper introduces DSI++, a continual learning challenge for Differentiable Search Indexes, along with the dataset for evaluation. The authors analyze the severity of this issue in FLAN-T5 model and locate this issue at memorization. The authors propose two techniques to tackle this problem in the context of continually indexing new documents. They propose a method for implicit forgetting based on Sharpness aware minimization and generate queries for old and new documents to use during continual indexing.

**Reasons To Accept:**

- The Differentiable Search Indices is a significant problem to work on. This paper is the first one to study the problem of updating DSI, which is an important feature of serving an LM for IR.

- This paper is well-written and well-presented.

- This work proposes two benchmarks built upon Natural Questions (NQ) and MS MARCO. These benchmarks could serve as good resources to foster relevant research in the community. And the authors promised to release these datasets in the Reproducibility Statement section.

**Reasons To Reject:**

- The experiments are only conducted on FLAN-T5 (if I didn't miss any experiments). It would be helpful if the authors introduced more experiments with different architectures to demonstrate the approaches are generalizable.

- The baseline methods seem missing. It would be helpful if the authors could add experiments with other continual learning methods, like LWF (Learning without Forgetting)

**Reproducibility:**

4: Could mostly reproduce the results, but there may be some variation because of sample variance or minor variations in their interpretation of the protocol or method.

**Reviewer Confidence:**

3: Pretty sure, but there's a chance I missed something. Although I have a good feel for this area in general, I did not carefully check the paper's details, e.g., the math, experimental design, or novelty.

---

> ### Author Rebuttal · Authors · 2023-08-29
>
> We are glad that the reviewer finds: (i) our paper to be **well-written and clear in terms of the presentation**, (ii) our work is the **first to study the vital problem of updating DSI** and appreciates our efforts in this direction, and (iii) **our proposed benchmarks are valuable** to the community.
>
> **[Additional architectures]**
>
> Our primary focus in this work is to examine the performance of DSI models when continually indexing new documents. Additionally, we investigate how the scale of the model, whether it's T5-base, T5-large, or T5-xl, impacts the phenomenon of forgetting during continual indexing. **While exploring the susceptibility of DSI-BART under continual indexing is an interesting question, we believe this aspect is tangential to our current study and leave it for future investigations, following previous work in this study in terms of model choice**.
>
> The original study by Tay et al. (2022), and subsequent research by Wang et al. (2022), employ a Transformer architecture for sequence-to-sequence learning. They initialize this Transformer architecture using pre-trained T5 checkpoints before memorizing or indexing documents. There has also been recent research by Bevilacqua et al. (2022), which uses a pre-trained BART checkpoint for initializing DSI and reported results comparable to the original DSI study. Therefore, **given considerations of computational cost, we adhere to the original work and initialize our DSI model with pre-trained T5 checkpoints**.
>
>
> **[Additional baselines]**
>
> There are several families of approaches for continual learning – **regularization-based, memory-based, optimization-based, and parameter-isolation-based approaches** (Sodhani et al. (2022)). In the related work section of our paper (Appendix B, lines 985-1140), we discuss most of these approaches in detail pointing out relevant baselines for DSI++ setup and comparing with them in Tables 1 and 3. The reviewer suggests comparing with the regularization-based approaches, like Elastic Weight Consolidation (EWC) and Learning without Forgetting (LwF). Regularization-based approaches alleviate forgetting by regularizing the objective such that it either penalizes the feature drift on already learned tasks (LwF) or discourages change in parameters that were important to solve previous tasks (EWC). However, these approaches are shown to underperform the vanilla episodic replay-based approaches on widely used continual learning benchmarks (Prabhu et al. (2020); Hussain et al. (2021)). Therefore, we compare our proposed approach with episodic memory (epsmem). Moreover, **in the context of DSI++, LwF would *require* queries corresponding to new documents to prevent forgetting the old documents**. However, according to the DSI++ problem setup (section 2.1, lines 138-170), **we do not have any queries related to new documents, thereby rendering the LwF method not applicable**.
>
> **We thank the reviewer for their valuable time in reviewing our paper. Should there be any remaining concerns, we are readily available to address them, and we sincerely hope that the reviewer might consider revising the scores accordingly.**
>
> [References]
>
> [Tay et. al (2022)] Tay, Yi, Vinh Tran, Mostafa Dehghani, Jianmo Ni, Dara Bahri, Harsh Mehta, Zhen Qin et al. "Transformer memory as a differentiable search index." Advances in Neural Information Processing Systems 35 (2022): 21831-21843.
>
> [Wang et. al (2022)] Wang, Yujing, Yingyan Hou, Haonan Wang, Ziming Miao, Shibin Wu, Qi Chen, Yuqing Xia et al. "A neural corpus indexer for document retrieval." Advances in Neural Information Processing Systems 35 (2022): 25600-25614.
>
> [Bevilacqua et. al (2022)] Bevilacqua, Michele, Giuseppe Ottaviano, Patrick Lewis, Scott Yih, Sebastian Riedel, and Fabio Petroni. "Autoregressive search engines: Generating substrings as document identifiers." Advances in Neural Information Processing Systems 35 (2022): 31668-31683.
>
> [Sodhani et. al (2022)] Sodhani, Shagun, Mojtaba Faramarzi, Sanket Vaibhav Meha, Pranshu Malviya, Mohamed Abdelsalam, Janarthanan Janarthanan, and Sarath Chandar. "An introduction to lifelong supervised learning." arXiv preprint arXiv:2207.04354 (2022).
>
> [Prabhu et. al (2020)] Prabhu, Ameya, Philip HS Torr, and Puneet K. Dokania. "Gdumb: A simple approach that questions our progress in continual learning." In Computer Vision–ECCV 2020: 16th European Conference, Glasgow, UK, August 23–28, 2020, Proceedings, Part II 16, pp. 524-540. Springer International Publishing, 2020.
>
> [Hussain et. al (2021)] Hussain, Aman, Nithin Holla, Pushkar Mishra, Helen Yannakoudakis, and Ekaterina Shutova. "Towards a robust experimental framework and benchmark for lifelong language learning." In Thirty-fifth Conference on Neural Information Processing Systems Datasets and Benchmarks Track (Round 1). 2021.

---

### Official Review · Reviewer_fVVm · 2023-08-10

**Soundness:** 3

**Excitement:**

3: Ambivalent: It has merits (e.g., it reports state-of-the-art results, the idea is nice), but there are key weaknesses (e.g., it describes incremental work), and it can significantly benefit from another round of revision. However, I won't object to accepting it if my co-reviewers champion it.

**Paper Topic And Main Contributions:**

The authors propose an approach based on Differentiable Search Indices (DSI) for lifelong learning paradigm, enhancing it with Sharpness-Aware Minimization and generative memory replay. Experimental results show the proposed approach can bring significant improvement of Hit@1/10 and index accuracy on previously learnt corpus, when the models are trained with a sequence of corpus one by one.

**Questions For The Authors:**

(A) The DSI paper [Tay et al.] utilizes three different kinds of NQ datasets, NQ10k, NQ100k and NQ320k. The author only examine their approach with NQ100k. Will the model fail to mitigate forgetting if dataset scale becomes larger?

(B) In standard lifelong learning paradigm, the model should not access previous datasets. In prior work about Episodic Replay (e.g., https://aclanthology.org/2020.emnlp-main.39.pdf), it is common to keep a tiny subset (1%) of previous datasets. However, the authors claim they use equal proportion of old and new corpora to obtain main results (of $cl(U_i)$). It is unclear whether SAM and ER or the high enough proportion of old examples play an role to reduce forgetting.

(C) The model trained with multi-task paradigm can usually be the upper bound of lifelong learning approaches based on Episodic Replay. However, the proposed approach in this work surpasses *from-scratch approach* but authors does not provide enough analysis about what content causes such forward-transfer.

(D) The model scale does affect the model's intrinsic ability to avoid forgetting. More results of larger models (like T5-XXL) are welcome.

**Reasons To Accept:**

1. Lifelong learning is a practical scenario and it is important to reduce catastrophic forgetting in lifelong learning. This work proves SAM and ER can help to reduce forgetting of indexing models.
2. The authors propose to split forgetting into two parts, implicit and explicit forgetting. And they empirically analyze the causes of both kinds of forgetting and discuss the approaches to mitigate them.
3. This paper is well organized and providing experimental evidence to support the effectiveness of DSI++.

**Reasons To Reject:**

1. This paper is not novel enough, since SAM and (generative) episodic replay have been proven to be effective to reduce catastrophic forgetting in many NLP tasks.
2. The authors examine the proposed approach within limited datasets and model architecture. As an improvement of DSI, the employed datasets in this work are not **the same** as DSI paper [Tay et al.]. The authors should explain why to modify dataset scale (See more in Question A).
3. There are unclear parts of experiment settings, which can lead to misleading or even wrong conclusions (See Question B-D).

**Reproducibility:**

3: Could reproduce the results with some difficulty. The settings of parameters are underspecified or subjectively determined; the training/evaluation data are not widely available.

**Reviewer Confidence:**

4: Quite sure. I tried to check the important points carefully. It's unlikely, though conceivable, that I missed something that should affect my ratings.

---

> ### Author Rebuttal · Authors · 2023-08-29
>
> We thank the reviewer for their time and remarks on our work. We are pleased that the reviewer finds the paper well-written, coherent, and comprehensive and appreciates our efforts in continually updating DSI.
>
> **[Novelty]**
>
> We list our contributions:
>
> 1. We **propose DSI++**, a continual learning challenge for the recently proposed Differentiable Search Indices (DSI) paradigm. To enable research progress on DSI++, **we introduce two benchmarks** based on the existing **Natural Questions** and **MS Marco** datasets. To understand the severity of the catastrophic forgetting, **we systematically study different design choices** like – document identifier representation strategies (unstructured atomic, naively structured, and semantically structured) and the pre-trained model size (T5-base, T5-large, T5-xl).
>
> 2. We are the **first to show that the DSI model undergoes a forgetting phenomenon during memorization**, leading to unstable learning dynamics. Mehta et al. (2021) show that explicitly optimizing for flatter loss basins using Sharpness-Aware Minimization (SAM) leads to lesser forgetting, especially in task-incremental learning settings where data undergoes a clear distributional shift. We extend this work to the new DSI paradigm and **convincingly demonstrate that SAM helps with the stable memorization of documents. Our results generalize the earlier findings even to the settings where data does not undergo a clear distributional shift (i.e., memorization task)**. Although SAM helps to stably memorize +12% of documents, there is still a lot of room for improvement and our work invites more future work in this direction.
>
> 3. We **propose a generative memory-based approach** to alleviate forgetting during continual indexing of the new documents. Note that in DSI++, we assume no access to ground truth queries for the older documents. Furthermore, we do not get to see queries for the new documents. Therefore, **we use generative memory to generate pseudo-queries for already indexed (older) documents as well as an incoming batch of new documents, thus, enabling us to leverage unlabeled data (in the form of new documents) for continual semi-supervised learning**. On the other hand, Sun et al. (2020) **assume that the incoming data are fully labeled, which is not applicable in DSI++** (we do not get to see queries for the new documents). Next, they learn a parametric model to reconstruct the examples for all of the seen tasks. However, such an approach strictly underperforms episodic memory which retains a subset of seen examples. In our work, we do not generate example pairs (document, queries) but rather generate pseudo-queries, and we show that our approach outperforms episodic memory that retains ground truth queries.
>
> We believe that the above list of contributions would make the paper’s novelty clearer to the reviewer. We have already discussed these contributions throughout our paper. With the extra page in the camera-ready version, we will explicitly enumerate the above list of contributions towards the end of the Introduction section for improved clarity.
>
> **[Question A: Scale of a data corpus]**
>
> To showcase that our proposed approach generalizes to datasets other than Natural Questions [NQ; Tay et al. (2022)], **we introduce one more benchmark for DSI++, constructed from a publicly available passage retrieval dataset, namely MS Marco** [Nguyen et al. (2016)]. We show that ER with generative memory improves the overall performance of the retrieval task, reducing forgetting of previously indexed documents and enabling forward transfer to newly indexed documents. We show that our results hold across two datasets – Natural Questions and MS MARCO, thus, showcasing the generalizability of our approach.
>
> To investigate the effectiveness of the generative memory with the scale of a corpus, we conducted experiments with a full MS MARCO passage retrieval dataset (≈ 8.9M passages). We construct two corpora – D$_0$ = 8M and D$_1$ = 841, 823 passages. We train the DSI model using D$_0$ passages and incrementally add D$_1$ passages. In Table 3 (in Appendix), we report results for MS MARCO. Overall, we show that our proposed solution convincingly reduces forgetting in large corpus settings. **In summary, we experiment with MS Marco 8.8M passages as opposed to NQ 320k (approx. 28 times larger corpus)**. We believe that the reviewer overlooked our detailed analysis of how our method interacts with the scale of a corpus, which we covered in Section 5.2 of our paper (refer to lines 584-602).
>
> **[Question B: Equal proportion of the old and new corpora]**
>
> Note that both the memorization and retrieval tasks return a document identifier corresponding to the input. So the DSI applications have access to the document contents for subsequent use. Therefore, by problem setup, we have access to the contents of the previous documents and one can continue indexing old and new documents with the hope of answering queries (retrieval task) for all seen documents. So we consider cl(U$_i$) as one of the baselines for our experimentation, where U$_i$ is all documents seen until the $i^{th}$ step. To update the DSI model with the indexing objective, **in the T5 tasks mixture**, we consider an equal proportion of the old and new documents. The hypothesis is that the model has already memorized the old documents, so aggressively memorizing the new documents 50% of the time and replaying older documents 50% of the time. **We want to clarify that the equal proportion relates to replay frequency, not a small subset of previous documents, as misinterpreted by the reviewer. We will revise the method description for clarity and to prevent future confusion**.
>
> Also, for the cl(U$_i$) baseline, we do not retain any queries corresponding to the older documents (i.e., 0% of previous queries). So there is no episodic replay for the retrieval task and as visualized in Figures 4, 7, and 8 (Tables 1, 3), cl(U$_i$) undergoes forgetting of the retrieval task.  On the other hand, from Figures 4, 7, and 8, we see **our proposed method [cl(U$_5$)+genmem(U$_5$)] outperforms cl(U$_5$) both in terms of average performance A$_n$ and forgetting F$_n$ over five sequential updates. These results convincingly show that the SAM and ER with generative memory significantly alleviate forgetting the retrieval task compared to considered baselines**.
>
> **[Question C: train-from-scratch baseline]**
>
> Multi-task learning, referred to as “train-from-scratch”, can serve as a performance upper-bound for lifelong learning when there is no detrimental interference among tasks, and all tasks are evenly balanced. However, in the case of DSI++, there exists an initial base corpus that is larger than subsequent corpora, leading to an imbalance among tasks. Consequently, **training from scratch should be regarded as a competitive baseline rather than an inherent upper bound**. This is also the reason we report the learning accuracy for every metric, which can be seen as an upper bound since it maintains a running average of the best performance across all corpora.
>
> Furthermore, one of the key objectives of lifelong learning systems is to leverage prior knowledge to enhance the learning of new tasks. Indeed, we observe that our proposed method excels in forward transfer compared to the train-from-scratch approach. For detailed results, please refer to Tables 1 and 3, specifically in the evaluation corpus D1 (right half of the table), and compare the rows labeled cl(U$_1$)+genmem(U$_1$) with those representing the train-from-scratch approach.
>
> **[Question D: Model scale]**
>
> In Section 2.4, we systematically study how the DSI model scale affects forgetting. (see Q5 — lines 242-243 and “Model scale.” paragraph — lines 289-299).
> From Figure 2, we observe that larger models outperform their smaller counterparts in terms of the average performance A$_n$ and the learning performance LA$_n$ (T5-xl > T5-large > T5-base). However, **empirically we report that forgetting F$_n$ is severe across all model scales, without any clear best performer**, and therefore given computational constraints, we focus on T5-base for the rest of our experimentation.
>
> **We express our gratitude to the reviewer for their thorough review and insightful questions. Please acknowledge if our responses adequately address your queries. We are available to provide further clarifications and eagerly await updates to the initial ratings.**
>
> [References]
>
> [Tay et al. (2022)] Tay, Yi, Vinh Tran, Mostafa Dehghani, Jianmo Ni, Dara Bahri, Harsh Mehta, Zhen Qin, et al. "Transformer memory as a differentiable search index." Advances in Neural Information Processing Systems 35 (2022): 21831-21843.
>
> [Mehta et al. (2021)] Mehta, Sanket Vaibhav, Darshan Patil, Sarath Chandar, and Emma Strubell. "An empirical investigation of the role of pre-training in lifelong learning." arXiv preprint arXiv:2112.09153 (2021).
>
> [Sun et al. (2020)] Sun, Fan-Keng, Cheng-Hao Ho, and Hung-Yi Lee. "LAMOL: LAnguage MOdeling for Lifelong Language Learning." In International Conference on Learning Representations. 2020.
>
> [Nguyen et al. (2016)] Nguyen, Tri, Mir Rosenberg, Xia Song, Jianfeng Gao, Saurabh Tiwary, Rangan Majumder, and Li Deng. "MS MARCO: A human generated machine reading comprehension dataset." In CoCo@ NIPs. 2016.

---

### Meta-Review · Area_Chair_wF3C · 2023-09-19

**Recommendation:** 5

**Metareview:**

The paper introduces DSI++, an approach using continual learning for Differentiable Search Indexes. The authors define a setting in which old and new queries are not available when updating a DSI with a new set of data, and they demonstrate that their proposed approach is able to mitigate the catastrophic forgetting happening on the old data when doing this operation. The reviewers all agree that the paper is well written and it discusses a topic worth of investigation to improve the capability of updating DSIs over time, which is a practical problem in real-world scenarios. The reviewers agree that the methodology is sound and that the provided experiments show its contribution.

---

### Decision · Program_Chairs · 2023-10-07

**Decision:**

Accept-Main

**Comment:**

The paper introduces DSI++, an approach using continual learning for Differentiable Search Indexes. The authors define a setting in which old and new queries are not available when updating a DSI with a new set of data, and they demonstrate that their proposed approach is able to mitigate the catastrophic forgetting happening on the old data when doing this operation. The reviewers all agree that the paper is well written and it discusses a topic worth of investigation to improve the capability of updating DSIs over time, which is a practical problem in real-world scenarios. The reviewers agree that the methodology is sound and that the provided experiments show its contribution.